# PREDICTING EVOLUTIONARY RATE AS A PRETRAINING TASK IMPROVES GENOME LANGUAGE MODELS

## ABSTRACT

Genome language models (gLM) have the potential to encode how and when genes are regulated without requiring labeled data. Most gLMs are pretrained using genome sequence reconstruction tasks inspired by natural language processing, such as masked language modeling (MLM) or next token prediction (NTP). Recent studies have shown that these gLMs often fail to capture biological signal, showing limited gains over simple classifiers on raw sequence or randomly initialized models on downstream genomic prediction tasks. To address these limitations, we explored alternative pretraining tasks for gLMs. Evolutionary rate has historically been the strongest predictor of function in genomics, but to date, there has been limited investigation of pretraining tasks exploiting evolution. Here, we introduce two evolution-based pretraining tasks that predict the rate of evolution from genomic sequence: current evolution prediction and masked evolution modeling. These tasks are designed so that they can be combined with NTP and MLM, enabling a systematic assessment of predicting sequence only, evolutionary rate only, or both. Using a novel suite of benchmarks that balance distinct aspects of genome function, we show that training on both sequence and evolutionary rate outperforms training on sequence alone. Moreover, for many tasks, training on evolutionary rate alone outperforms training on sequence alone. These results demonstrate that evolution-based pretraining offers a principled alternative or additional task to sequence reconstruction, establishing evolution as a key training target for genome-scale models.

## 1 INTRODUCTION

Genome language models (gLMs) are models pretrained in a self-supervised manner on genomic sequences. Among other applications, gLMs promise to learn a foundational understanding of genomic "grammar", or how and when genes are activated, repressed, or modulated by their regulatory context. Learning this grammar would be particularly impactful for human gene regulation, where our understanding remains incomplete due to the complexity of eukaryotic genomic architectures, enabling advances in personalized medicine, drug discovery, synthetic biology, and related areas. For this reason, since the first gLMs were proposed in 2021, there has been a rapid increase in the number of proposed models, and the financial scale of these efforts has also grown (Dalla-Torre et al., 2025; Nguyen et al., 2023; Ji et al., 2021; Schiff et al., 2024; Benegas et al., 2024; Albors et al., 2025; Brixi et al., 2025). For example, the Arc Institute used over 2,000 NVIDIA H100 GPUs to train Evo2, a gLM trained on over 9 trillion nucleotides from thousands of species [1]

However, there has been little exploration of the pretraining tasks used to train gLMs. Most of these models are pretrained using sequence reconstruction tasks inspired by natural language processing such as predicting the next token given previous tokens (next token prediction (NTP)) (Figure 1A) or predicting masked tokens from surrounding context (masked language modeling (MLM)) ((Figure 1B). While these models have demonstrated strong performance after fine-tuning on tasks like predicting transcription factor binding, chromatin profile activity, and regulatory element identification (Dalla-Torre et al., 2025), recent works have exposed that they underperform simple baselines in the zero-shot setting (Marin et al., 2023; Tang et al., 2025) and cannot reliably outperform randomly-initialized versions of their architectures (Vishniakov et al., 2024). Zero-shot performance is partic-

---

[1]Based on training described at https://arcinstitute.org/news/evo2

ularly important in the context of biological understanding, where labels may not be known *a priori*. Current zero-shot gLM performance suggests that current pretraining tasks may not strongly learn genomic grammar; instead, gLM performance may be driven by technical factors such as overparameterization, a phenomenon previously reported for protein language models (Li et al., 2024)

In contrast to gLM pretraining, which relies solely on sequence reconstruction, many bioinformatic methods for inferring genome function exploit comparative genomics. Comparative genomics uses cross-species genome comparisons to identify the function of genomic elements through patterns of shared ancestry. Large-scale repositories have sequences for thousands of species' genomes (Lewin et al., 2018), enabling the construction of whole genome alignments against the human genome. These alignments allow the direct estimation of evolutionary rates at the single-base resolution. Estimating evolutionary rate is useful because it correlates with functional importance: functional sites tend to be conserved, meaning they accumulate substitutions more slowly than neutrally evolving ones. Crucially, evolutionary rate is a highly compressed summary of an MSA that may span hundreds of genomes, yet preserves the essential signal of functional constraint. For this reason, evolutionary rate remains one of the most powerful and widely applied predictors of genome function today (Zhou & Troyanskaya, 2015; Consens et al., 2025; Pollard et al., 2010; Rentzsch et al., 2019; Zhou et al., 2011; Benegas et al., 2024; Albors et al., 2025). Despite its effectiveness in detecting function, evolutionary rate has not yet been widely investigated for training gLMs.

In this work, we provide a principled study of the effectiveness of predicting evolutionary rate as a pretraining task. We introduce two novel evolutionary rate prediction pre-training tasks (Figure 1). In current evolution prediction (CEP), the model learns to predict the evolutionary rate at each position given the sequence up to that position (Figure 1C). In masked evolution modeling (MEM), the model learns to predict evolution rates at masked positions from the surrounding nucleotides (Figure 1D). Notably, these tasks are compatible with NTP and MLM, respectively, and only require single sequences as inputs (Figure 1E-F). We use these tasks to study the effectiveness of evolutionary-rate pretraining, by training evolutionary-rate aware gLMs that we call "Gamba" models. Critically, we also train models, using the same architectures and training data, with sequence reconstruction tasks, and with both evolutionary-rate and sequence reconstruction, allowing us to directly compare the pretraining strategies and to evaluate if they can synergize with each other. We developed a biologically-aligned zero-shot benchmark, and we show that incorporating explicit modeling of evolutionary rate through CEP or MEM consistently improves representation quality. In particular, adding CEP or MEM to NTP or MLM yields up to >13% gains in balanced accuracy on regulatory element classification over sequence-only pretraining. Notably, in some cases MEM-only training outperforms MLM alone, highlighting the effectiveness of evolutionary rate prediction as a standalone pretraining signal composable with sequence reconstruction.

## 2 RELATED WORKS

### 2.1 TRAINING GLMS WITH EVOLUTION

In principle, pretraining with multi-species sequence reconstruction implicitly captures evolutionary constraints. By encountering homologous genomic regions across species, gLMs can learn to conserved patterns that reflect functional elements, analoguous to how protein language models trained on single sequences from across species capture functional and structural constraints (Zhang et al., 2024). Indeed, pretraining on evolutionarily diverse sequences improves the capabilities of gLMs in modeling functional relevance (Dalla-Torre et al., 2025; Consens et al., 2025) as opposed to human-only pretraining. However, genomes are significantly noisier than protein sequences, and are dominated by "junk DNA" of unclear functional significance (Fagundes et al., 2022). It is unclear whether presenting models with genomes across species is as effective a strategy for implicitly learning evolution in genomes as it has been for proteins. Furthermore, genomes are much longer than proteins, making it more enticing to compress evolutionary information from related species into a simple target, and unlike in protein modeling, we are often primarily interested in the human genome. Thus, in our work, we assess the effectiveness of directly predicting evolution.

One way to directly incorporate evolutionary signal **explicitly** during pretraining is by reconstructing MSAs instead of single sequences. GPN-MSA is a masked language model over MSAs (Benegas et al., 2024) with strong performance on noncoding variant effect prediction tasks. GPN-STAR improves performance on noncoding variant effect prediction further by using the phylogenetic tree

inferred from the MSA to constrain attention between sequences (Ye et al., 2025). However, requiring MSAs as inputs limits downstream applications (e.g., synthetic sequences, which will not have natural homologs in other species). The closest approach to ours is PhyloGPN, which also predicts evolutionary rate as a target (Albors et al., 2025). However, instead of predicting the rate at every position (CEP) or at many masked positions (MEM), PhyloGPN predicts the rate at a single central nucleotide per input. This is less efficient to train, and prior work also does not attempt to evaluate if evolutionary rate-based pretraining can be composed with sequence reconstruction. In contrast, we combine CEP with NTP and MEM with MLM, allowing us to conduct ablations using the same architecture and training data to quantify the impact of training on evolutionary rate only, sequence only, or both.

## 2.2 BIOINFORMATICS BASELINES INCORPORATING EVOLUTION

Bioinformatics baselines estimating evolutionary rate such as PhyloP scores, PHAST scores, and CADD predictions often outperform state-of-the-art gLMs on zero-shot tasks such as variant effect prediction (Pollard et al., 2010; Brixi et al., 2025; Zhou et al., 2011; Rentzsch et al., 2019). The success of these scores in variant effect prediction suggests evolutionary rate may be an effective pretraining signal for gLMs. Yet, to date, there has been little systematic evaluation of how this signal compares to or can be combined with sequence-only pretraining tasks, as opposed to using these methods as stand-alone baselines for gLMs.

## 2.3 EVALUATION SUITES FOR GLMS

A key challenge for genome language models is evaluating whether pretraining captures biologically meaningful signals across the diversity of genomic elements. Genomic elements operate under different mechanisms, have varying degrees of knowledge and annotation associated with them, and vary in relative abundance in the genome. An effective benchmark for assessing performance of pre-trained gLMs should have coverage over all of these factors, while enabling fair comparisons between models without requiring excessive compute. Existing benchmarks have made important progress but typically address only part of this space: for example, many focus on binary classification of singular elements in isolation, and few include rare elements such as ultra-conserved noncoding elements (UCNEs) (Visel et al., 2007; Grešová et al., 2023; Marin et al., 2023; Zhou et al., 2023; Dimitrieva & Bucher, 2013). In addition, many evaluations require training auxiliary models on top of frozen embeddings, which complicates cross-model comparison and adds computational cost (Tang et al., 2025; Marin et al., 2023). Randomly initialized baselines are also rarely reported, making it difficult to deconvolve the effects of architecture and size from those of pretraining (Vishniakov et al., 2024). To address these gaps, we introduce a benchmark suite that spans distinct genomic elements of different regulatory mechanism, prior knowledge, and relative abundance, supports multi-class discrimination, enables efficient zero-shot evaluation, and includes randomly initialized baselines for fair comparison.

## 3 METHODS

### 3.1 CURRENT EVOLUTION PREDICTION AND MASKED EVOLUTION MODELING

We propose two pretraining tasks that predict evolutionary rate given genomic sequence as input: current evolution prediction (CEP) and masked evolution modeling (MEM) (Figure 1). While a variety of scores estimate evolutionary rate, we used PhyloP scores (Pollard et al., 2010), a widely-adopted score (Perez et al., 2025) in phylogenetics that measures evolutionary conservation and acceleration at a single-nucleotide resolution. PhyloP scores quantify whether a nucleotide position is evolving slower (conserved; positive score) or more rapidly (accelerated; negative score) than expected compared to a neutral evolutionary model, and they are computed from a multiple sequence alignment (MSA) along with its corresponding phylogenetic tree.

While PhyloP scores are calculated from an MSA, we pretrained our models to predict PhyloP scores from single unaligned sequences. CEP and MEM predict PhyloP scores from different sequence contexts, paralleling the dominant sequence reconstructing pretraining tasks for gLMs. First, CEP predicts the PhyloP score at the $i$-th position using only the sequence context up to and including

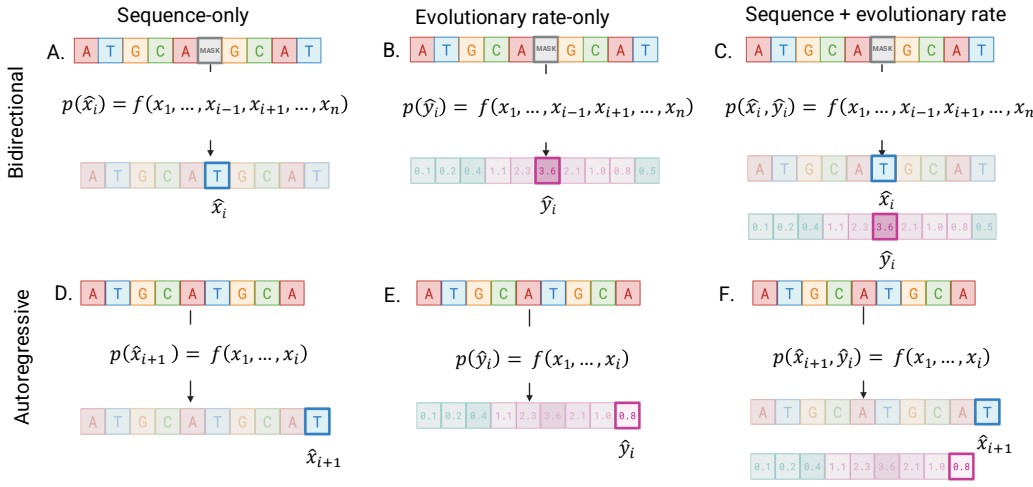

Figure 1: Comparison of evolutionary-rate based and existing sequence-only pretraining methods. A-C show tasks with bidirectional sequence context, while D-F show tasks with autoregressive context. A) Masked language model (MLM). B) Masked evolution modeling (MEM). C) MLM combined with MEM. D) Next token prediction (NTP). E) Current evolution prediction (CEP) F) NTP combined with CEP.

that position. Formally, given a sequence $x = x_1, x_2, ..., x_i$, the model outputs a predicted PhyloP score $\hat{y}_i$ at the $i$-th position. This setup mimics next token prediction, except instead of predicting the "next" token $i + 1$, the current rate at $i$ is predicted (Figure 1C). Second, MEM predicts PhyloP scores given bidirectional context. During training, a subset of positions in the input sequence (15%) is randomly selected and masked. The model is then trained to reconstruct the PhyloP scores at these masked positions, using the entire sequence as context (Figure 1D). This allows the model to leverage bidirectional context from both upstream and downstream sequence.

Because CEP and MEM operate on the same inputs used for NTP and MLM, respectively, our conservation prediction tasks can be combined with sequence reconstruction tasks. CEP can be combined with NTP in the autoregressive setting (Figure 1E), and MEM can be combined with MLM in the bidirectional setting (Figure 1F).

### 3.1.1 LOSS FUNCTIONS

Gamba models predict a mean and log variance for the evolutionary rate at each position, and are trained with a Gaussian negative log likelihood (GNLL) loss:

$$\mathcal{L}_{\text{GNLL}} = \frac{1}{N} \sum_{i=1}^{N} \left[ \frac{1}{2} \left( \log\big(\max(\sigma_i^2, \epsilon)\big) + \frac{(y_i - \mu_i)^2}{\max(\sigma_i^2, \epsilon)} \right) \right] + \text{const.}$$

where $y_i$ is the true conservation score, $\mu_i$ is the predicted mean, $\log \sigma_i^2$ is the predicted log variance and $N$ is the number of supervised positions in the batch. For the sequence prediction tasks (NTP and MLM), we used a standard cross entropy (CE) loss on the predicted sequence logits over the 4 nucleotides (A, T, C, and G), which seeks to maximize the predicted likelihood of the true nucleotide at each position. We combined evolutionary rate prediction and sequence reconstruction by simply adding the two losses: $L = L_{\text{GNLL}} + L_{\text{CE}}$.

### 3.2 THE GAMBA MODEL ARCHITECTURES

The autoregressive Gamba (ArGamba) models are based on the Jamba architecture, a hybrid Transformer–Mamba model with mixture-of-experts (MoE) routing (Lieber et al., 2024). Jamba interleaves Transformer (Vaswani et al., 2017) and Mamba (Gu & Dao, 2023) blocks to combine the global modeling capacity of attention with the efficient long-range processing of state space models.

The bidirectional Gamba (BiGamba) models are based on the Caduceus architecture (Schiff et al., 2024). Caduceus is a bidirectional, Mamba state space model adapted to be reverse-complement equivariant, an inductive bias that captures the double-stranded nature of DNA. For both Gamba models, we evaluated three pretraining tasks:

- **Sequence-only**: Models trained to reconstruct genomic sequences only. For ArGamba, we trained models with NTP; and for BiGamba, we trained models with MLM.
- **Evolution-only**: Models trained to predict PhyloP score only. For ArGamba, we trained models with CEP; and for BiGamba, we trained models with MEM.
- **Sequence and evolution**: Models trained to both reconstruct sequence and predict PhyloP scores. For ArGamba, this is NTP + CEP; and for BiGamba, this is MLM + MEM.

For all PhyloP-score prediction tasks, the conservation head consists of a linear layer that maps the final hidden representation at each position to a 2-dimensional output: the predicted mean and log variance of a Gaussian distribution over conservation scores. For all sequence reconstruction tasks, a linear head maps the final hidden state to the nucleotide vocabulary. Models trained on both tasks use both heads.

### 3.3 PRETRAINING DATA

We trained Gamba on the human reference genome (hg38), obtained from Basenji (link) (Kelley et al., 2018). We excluded repetitive and structurally ambiguous regions using RepeatMasker annotations (Fernandes et al., 2020) and low-quality read centromere data, both obtained from the UCSC genome browser (Perez et al., 2025). This is consistent with recent gLMs such as GPN-MSA and Evo2 that address signal-to-noise ratio in the genome by adopting structured training regimes that over-sample informative regions while down-weighting repetitive elements (Benegas et al., 2024; Brixi et al., 2025), but we exclude repetitive and low-quality regions entirely to improve computational efficiency for our experiments. Chromosomes 3 and 16 are held out for validation, while chromosomes 2 and 22 are reserved for testing. We further exclude regions containing more than 10% ambiguous bases ("N") within any 1000 bp window. After filtering, the training corpus is reduced from ∼3 billion base pairs (bp) to ∼ 1 billion bp.

For evolutionary rate prediction, we processed PhyloP scores from the Zoonomia 241-mammal alignment (Consortium, 2020). The complete hg38 human reference genome is initialized with zeros, and true PhyloP values are indexed where available; positions without a score remain zero. All values are rounded to two decimal places.

## 4 RESULTS

To assess whether pretraining by predicting evolutionary rate improves gLMs, we evaluated models across three biologically aligned zero-shot tasks, two of which are new. We compare ArGamba and BiGamba against several representative gLMs. The Nucleotide Transformer suite provides both a human-reference-only model and a multi-species model (Nguyen et al., 2023), enabling assessment of whether evolutionary information can be captured implicitly by training on multi-species genomes and how this compares against our strategy (Dalla-Torre et al., 2025). To match ArGamba pretraining, we included HyenaDNA, which also uses single-nucleotide tokenization and autoregressive training. To match BiGamba pretraining, we included Caduceus, which is trained on the human reference genome alone (Schiff et al., 2024). We also report results for PhyloGPN, which explicitly predicts evolutionary signals from sequence and is therefore the most comparable to our evolutionary rate-based pretraining tasks (Albors et al., 2025). For model context lengths and parameter sizes see Supplementary Table A1.

### 4.1 PREDICTING EVOLUTIONARY RATE AS A PRETRAINING TASK IMPROVES REPRESENTATIONS

We designed a zero-shot evaluation to measure whether pretrained gLM representations can separate different genomic elements within the genome. Specifically, we evaluated the ability of a model's representation space to separate different categories of genomic regions with varying complexity and

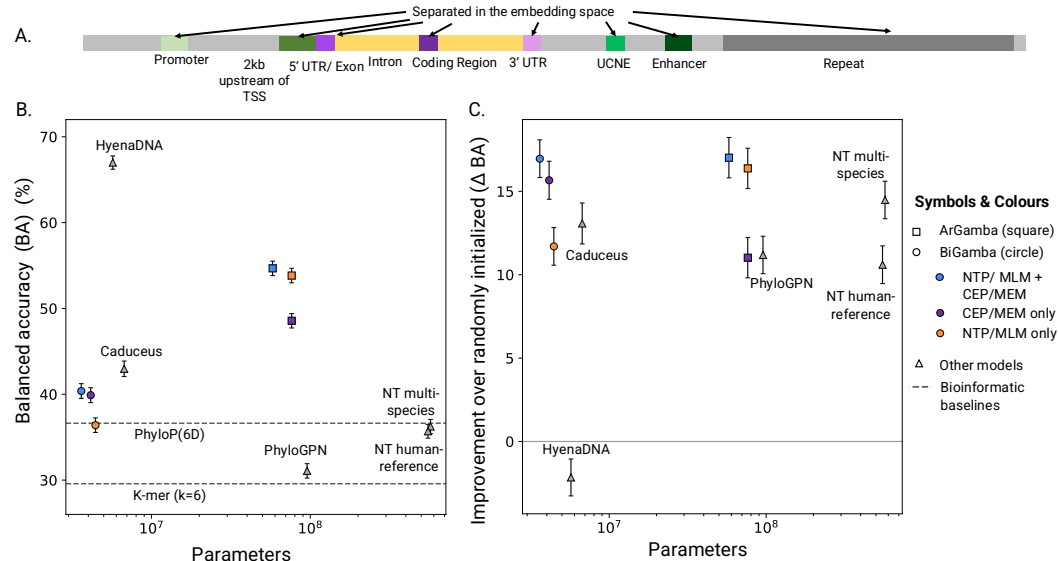

Figure 2: The global representation task. A) Examples of genomic categories. B) balanced accuracy by model size. C) Gain in balanced accuracy via pretraining by model size.

rarity (enhancers, UCNE, repeats, exons, introns, noncoding regions, coding regions, 2kb upstream of the transcription start site (TSS), 5'UTR, 3'UTR, promoters) (Supplementary A).

For this task, sequences of up to 2048 bp for Gamba, 6000 bp for the Nucleotide Transformer models, 131k bp for Caduceus, and 161k bp for HyenaDNA, were extracted with the region of interest (ROI) corresponding to the genomic element placed at the center (MLM-style models) or end (autoregressive models). Embeddings were obtained from the final hidden layer, and separation was evaluated using leave-one-out cross-validation with a 1-nearest neighbor classifier, which measures if genomic elements are clustered in embedding space (Figure 2A, Table A2).

We also report the 1-nearest neighbour classifier results for two non-gLM baselines, one sequence-based and one evolutionary rate-based. First, the sequence-based baseline evaluates if pretrained representations can outperform raw sequence alone. Because our downstream tasks require comparing sequences of different lengths, we implemented a $k$-mer model with $k = 6$. The evolutionary rate baseline evaluates whether pretraining on evolutionary rate can outperform using evolutionary rate directly. To produce sequence-level vectors of evolutionary rate scores, we summarized phyloP scores into six features: mean, variance, and the counts and means of positive and negative positions.

Pretraining tasks that combine evolutionary rate and sequence prediction provided the largest gains in accuracy compared to randomly initialized versions of the models. While HyenaDNA achieved the highest accuracy on this task, this performance appears to be largely due to inductive bias, as pretraining reduces the performance of the randomly initialized model (a phenomenon for this model also previously documented by Vishniakov et al. (2024)). Therefore, we focus on improvement over random initialization to disentangle the effects of pretraining from architecture. The largest improvements are achieved by ArGamba NTP+CEP and BiGamba MLM+MEM, with more than 16.5% improvement over random initialization (Figure 2B, Table A2). BiGamba MEM-only outperforms BiGamba MLM-only, suggesting that, in some cases, evolutionary rate alone can provide a stronger training signal than sequence alone. This effect is not observed for ArGamba NTP-only vs. ArGamba CEP-only, implying that it may depend on bidirectional context.

BiGamba MLM+MEM performs comparably to Caduceus (40.37% compared to 42.98%) despite using half the parameters and a much shorter context length (2048 bp vs. 131,000 bp). Finally, while we observe that training Nucleotide Transformer on multiple species (in their work, 850 genomes) over the human genome alone induces an improvement on representing genomic elements, similar to the improvement that our evolutionary rate pretraining tasks induce over training on human genome

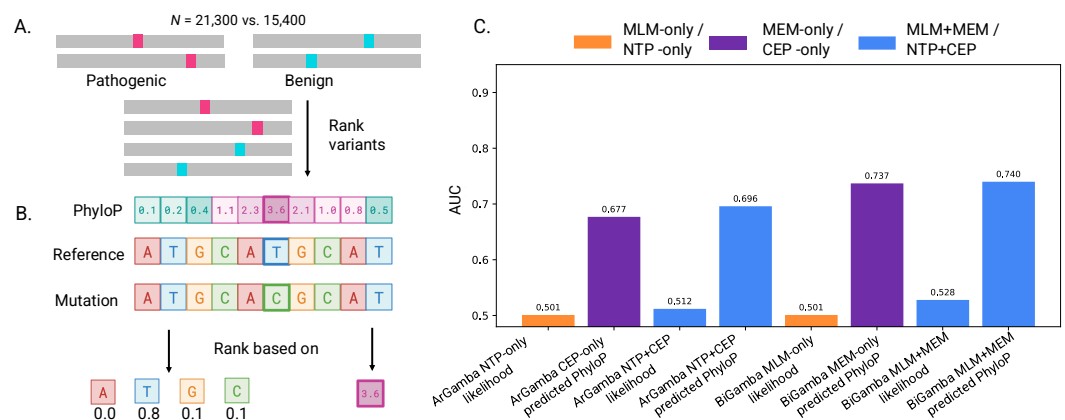

Figure 3: Variant effect prediction. A) The variant effect prediction task. B) Scoring and ranking variants. C) Gamba performance on the ClinVar dataset.

sequence alone, our proposed task does not increase the size of the training data corpus, making it more compute efficient. Together, these comparisons underscore the efficiency of evolutionary rate-based objectives.

## 4.2 PREDICTING EVOLUTION IMPROVES VARIANT EFFECT PREDICTION

However, the genomic categories in our previous benchmark provide only a coarse-grained view of function, while researchers are often interested in fine-grained effects (e.g., distinguishing between different kinds of functions mediated by a genomic element, as opposed to just identifying the broader type of genomic element). To evaluate fine-grained predictive ability, we tested variant effect prediction (VEP) (Zhou & Troyanskaya, 2015), an established benchmark that ranks genomic sequences based upon their pathogenicity, or their propensity to cause disease (Figure 3A). Previous works have employed the ClinVar dataset for this purpose, as it curates benign and pathogenic variants of human genomic sequences (Benegas et al., 2024; Landrum et al., 2014).

We rank variants in two ways (Figure 3B). First, as done by GPN-MSA (Benegas et al., 2024), variant effect scores are taken as the ratio of the likelihoods of the reference nucleotide and the mutated nucleotide, averaged across the forward and reverse strands. Second, we explore using PhyloP predictions instead of sequence to rank variants. For Gamba models also trained to predict the evolutionary rate, we used the average predicted PhyloP score of the forward and reverse strands. This strategy is consistent with previous work that uses PhyloP scores to predict pathogenic variants (Brixi et al., 2025; Benegas et al., 2024; Pollard et al., 2010).

Models trained to predict evolutionary rate consistently outperform sequence-only models (Table A4). While sequence log-likelihood AUCs for all models remain close to random (except the Caduceus model, performance previously reported by Albors et al. (2025)), adding CEP or MEM as a task in pretraining improves predictive power using sequence. In contrast, using predicted PhyloP scores yields stronger performance than sequence likelihood across MEM-only, MLM+MEM, CEP-only, and NTP+CEP models (Figure 3C, Table A4). Joint training on sequence and PhyloP scores provides the best discrimination of benign versus pathogenic variants. For example, ArGamba NTP+CEP outperforms ArGamba CEP-only in both predicted PhyloP AUC (0.696 vs. 0.677) and correlation (0.398 vs. 0.358), suggesting that sequence prediction enhances evolutionary rate learning. A similar trend is observed with BiGamba.

Among the neural network models we trained, BiGamba MLM+MEM performs best, with a predicted PhyloP score AUC of 0.7397 and a Pearson correlation of 0.4628 to true PhyloP scores. Although true PhyloP scores remain the strongest single predictor (AUC 0.9121), gLMs explicitly pretrained on evolutionary rate (MEM or CEP) substantially close the gap relative to sequence-only models.

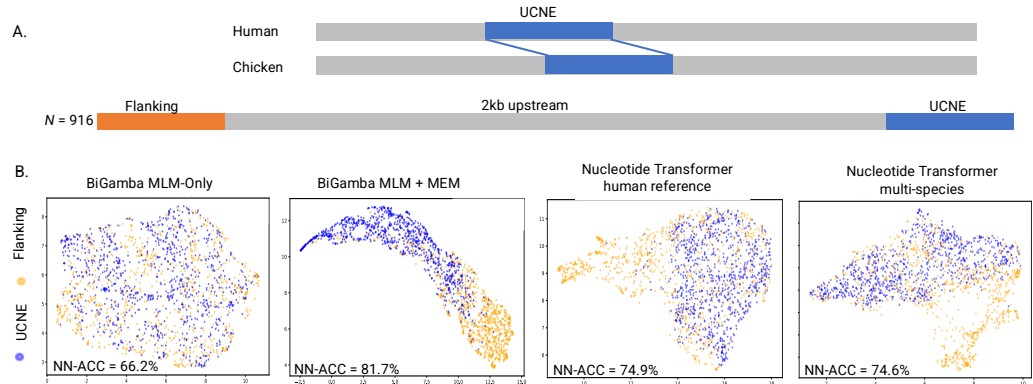

Figure 4: UCNE detection. A) UCNEs and flanking regions. B) UMAPs of pretrained representations of UCNEs and their flanking regions.

### 4.3 IDENTIFICATION OF ULTRACONSERVED NONCODING ELEMENTS SIGNIFICANTLY BENEFITS FROM EXPLICIT PRETRAINING ON EVOLUTION

Finally, we evaluated whether gLMs can identify ultraconserved noncoding elements (UCNEs) from sequence alone. UCNEs are rare noncoding regions that are highly conserved across very diverged species, specifically defined as noncoding regions longer than 200 bp with ≥95% sequence identity between humans and chickens (Dimitrieva & Bucher, 2013) (Figure 4A). Despite their extreme conservation, their biological roles remain poorly understood, and to date, sequence-based determinants of UCNEs have not been identified, meaning that UCNEs are generally only identified through MSAs, not single sequences. Hence, we reasoned that successful discrimination of UCNEs from single sequences would provide a test of whether gLMs can reveal insight into novel biology.

We evaluated each model's ability to discriminate between ultraconserved noncoding elements (UCNEs) and their flanking regions 2kb upstream. We chose to use the flanking regions of UCNEs as our negative class as we expect these regions to share sequence characteristics that are not determinants of UCNEs due to their shared genomic context (e.g., GC content). We also experimented with exons as our negative class, which like UCNEs are also highly conserved, which yielded similar results in the ranking of models as using flanking context (Supplementary Table **??**). We evaluated the embeddings from each model on binary classification between UCNEs and flanking regions using leave-one-out cross-validation with a 1-nearest neighbor classifier. We only evaluate UCNEs that have no homologs, and are found in the held out chromosomes of our datasets ($n = 916$). Therefore for the Gamba models, evaluation is restricted to the held-out set; for all other models, we evaluate the same UCNEs which may or may not be held out (as each model is trained using their own unique data split).

As shown in Table A3 and Figure 4B, models pretrained to predict evolutionary rate consistently outperform their sequence-only counterparts. ArGamba NTP+CEP achieves 77.6% balanced accuracy, a +13.7% improvement over ArGamba NTP-only; while BiGamba MLM+MEM reaches the best overall accuracy of 81.7%, a +15.5% gain over BiGamba MLM-only. Nucleotide Transformer human-reference and Nucleotide Transformer multi-species perform comparably, suggesting the gains in accuracy in separating UCNEs from flanking regions is specific to *explicitly* modeling evolutionary rate (Figure 4B). These results indicate that explicit evolutionary rate prediction provides sharper contrast between UCNEs and surrounding non-functional regions. Models trained with evolutionary rate-based pretraining (CEP-only, MEM-only) perform competitively but fall short of their hybrid counterparts, highlighting that the strongest performance arises from jointly modeling sequence and evolutionary rate.

## 5    Conclusion

We demonstrated that predicting evolutionary rates from the nucleotide sequence improves the performance of genome language model representations. We introduced two evolutionary rate-based pretraining tasks, used them to train the Gamba family of models, and demonstrated their advantages over traditional sequence reconstruction tasks. Our results show that evolutionary rate-only models outperform sequence-only baselines on several benchmarks, underscoring the strength of evolutionary signal as a training target.

Taken together, our findings establish evolution as a key training signal for gLMs and point toward pretraining strategies that more closely align model learning with the underlying biology of the genome. Our results suggest gLMs could further benefit from incorporating additional comparative genomics signals into self-supervised training. Although in our work, we use PhyloP scores derived from 240 mammalian species from the Zoonomia project (Consortium, 2020), future would could experiment with alternative scores, including PhastCons scores (Zhou et al., 2011), which capture evolutionary rates over broader contexts, and phyloP scores from alternative MSAs, such as vertebrate alignments, which have been shown to outperform the 240-mammalian–derived PhyloP scores on variant effect prediction (Benegas et al., 2024), possibly because they even further expand the diversity of species taken into account for the PhyloP score calculation.

Our study has several key limitations. First, as we were primarily interested in studying the value of evolutionary rate as a pretraining task, we trained models with much fewer parameters, lower context sizes, and less training data (e.g. by removing repeats instead of downweighing them) than current SOTA gLMs. This means that the Gamba models we present in this work are likely not as performant as they could be, and future work in scaling them is necessary to make them more competitive in downstream applications. Second, several design decisions could make our proposed pretraining tasks more effective. While we predict evolutionary rate only at masked positions in our MEM task, which we chose to do to make it more similar to MLM, it would in principle be more efficient to predict evolutionary rate at all positions. Similarly, we used a balanced loss between sequence reconstruction and evolutionary rate prediction, but future work could optimize parameters trading off between the losses.

In sum, we demonstrate the potential of combining comparative genomics and deep learning to understand genomes. Despite being a fundamental strategy for many bioinformatics methods used to understand genomes, comparative genomics has not yet been thoroughly investigated for training gLMs. By demonstrating that predicting evolutionary rate can be combined with sequence reconstruction strategies currently employed to train gLMs to further learn signal, we hope to spur future investigation in incorporating evolutionary signal into gLMs.

## 6    Resource Availability

Code and model weights are made available at (links redacted for double-blind review).

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

## A   Appendix

### Genomic Region Selection

We assembled a diverse set of genomic categories from established resources to evaluate model performance across simple, complex, and rare regulatory elements.

Promoters are regulatory regions in the non-coding genome that serve as the binding site for the transcriptional protein complexes that initiate transcription of a gene from DNA to RNA. Promoter annotations were obtained from the Eukaryotic Promoter Database (EPD) (Périer et al., 2000).

Enhancers are non-coding regulatory regions that increase gene expression, by increasing the likelihood of transcription, typically in a manner that depends on cell and tissue context (Visel et al., 2007). Enhancers were sourced from the VISTA Enhancer Browser (Visel et al., 2007), which experimentally validate putative enhancers in developing mouse embryos as driving gene expression in specific tissues.

Ultra-conserved noncoding elements (UCNEs) were taken from UCNEbase (Dimitrieva & Bucher, 2013), and are defined as sets of non-coding regions in the genome longer than 200 bp with $\geq 95\%$ sequence identity between humans and chickens.

Repeats encompass a broad class of genomic elements, short or long, that appear multiple times in the genome, or otherwise are considered low complexity DNA regions (Fernandes et al., 2020). In some cases, repeats are caused by transposable elements, mobile genetic elements that can move themselves and duplicate in the genome despite being largely non-functional for the host species. Repeats were extracted from the UCSC RepeatMasker track (Fernandes et al., 2020).

Protein-coding exons, introns, upstream transcription start site (2kb upstream of TSS), 5'UTRs, 3'UTRs, and coding versus noncoding regions were derived from the GENCODE human genome annotations (GTF format) (Frankish et al., 2021). All regions from the GENCODE human genome annotations were filtered to canonical transcripts: a single, representative transcript identified at every gene (usually with the highest coverage of conserved exons, highest expression, longest coding sequence, etc.) (Dyer et al., 2025). Introns are regions within genes that are spliced out, and do not encode for proteins, but can play important roles in mRNA processing. Exons are regions within genes that remain after splicing, but can be non-coding, meaning they do not get translated to proteins, if they are UTRs. 5'UTRs and 3'UTRs are specific classes of exons involved in splicing and regulation of translation as well as RNA stability. The region defined as 2kb upstream of the Transcription Start Site (TSS) is considered a functionally rich region which may contain sequences involved in transcription initiation, including promoters.

### ArGamba & BiGamba Training

Models were trained for roughly seven epochs on either a single NVIDIA L40S (48GB) or NVIDIA RTX A6000 (48GB). We used the Adam optimizer with a learning rate schedule defined by a linear warmup followed by inverse square root decay, implemented via a LambdaLR scheduler.

### Training Data

Chromosome sizes and centromere annotations were downloaded from the UCSC Genome Browser (Perez et al., 2025). The full human genome sequence (hg38.ml.fa) was obtained from the Basenji Barnyard resource (Kelley et al., 2018), and repeat elements were collected from the UCSC Repeat-Masker track (Fernandes et al., 2020). PhyloP scores were downloaded from the 241-mammalian alignment hub provided by the Comparative Genomics Lab at UCSC (Pollard et al., 2010). Chromosomes 2 and 22 were held out for test, and chromosomes 16 and 3 for validation.

### Supplementary Figures & Tables

Table A1: All models evaluated with parameter counts (in millions) and context lengths (in 1,000 bp).

| Model | Params (M) | Context (kbp) |
|---|---|---|
| ARGAMBA NTP-ONLY | 66.5 | 2 |
| ARGAMBA CEP-ONLY | 66.5 | 2 |
| ARGAMBA NTP+CEP | 66.5 | 2 |
| BIGAMBA NTP-ONLY | 3.9 | 2 |
| BIGAMBA MEM-ONLY | 3.9 | 2 |
| BIGAMBA MLM+MEM | 3.9 | 2 |
| NT multi-species | 498.3 | 6 |
| NT human-ref | 480.4 | 6 |
| PhyloGPN | 83.2 | 0.481 |
| HyenaDNA | 6.6 | 160 |
| Caduceus | 7.7 | 131 |
| K-mer (k=6) | 0.0 | 2 |
| PhyloP (6D) | 0.0 | 2 |

Table A2: Representation performance (balanced accuracy) across genomic categories (vista enhancer, UCNE, repeats, exons, introns, noncoding regions, coding regions, 2kb upstream of the TSS, 5'UTR, 3'UTR, promoters) sampled from the whole genome. The final column reports relative improvement over random initialization. Random initialization baseline follows Vishniakov et al. (2024). Models in blue are our models pretrained to predict PhyloP scores, models in orange are our models pretrained on sequence alone, models in purple are pretrained on both tasks. Bolded entries in the pretrained column indicate the best performing model. Bolded entries in the Improvement column indicate the largest relative performance increase.

| Model | Random Init (%) | Pretrained (%) | Improvement (+%) |
|---|---|---|---|
| NUCLEOTIDE TRANSFORMER (HUMAN-REF) | 25.09 | 35.69 | +10.60 |
| NUCLEOTIDE TRANSFORMER (MULTI-SPECIES) | 21.76 | 36.24 | +14.48 |
| HYENADNA | 69.15 | **66.99** | -2.16 |
| PHYLOGPN | 19.89 | 31.07 | +11.18 |
| CADUCEUS | 29.89 | 42.97 | +13.08 |
| GAMBA NTP-ONLY | 37.46 | 53.83 | +16.37 |
| GAMBA CEP-ONLY | 37.54 | 48.56 | +11.02 |
| GAMBA NTP+CEP | 37.65 | **54.67** | **+17.02** |
| BI-GAMBA MLM-ONLY | 24.70 | 36.40 | +11.70 |
| BI-GAMBA MEM-ONLY | 24.23 | 39.90 | +15.66 |
| BI-GAMBA MLM+MEM | 23.42 | 40.37 | +16.96 |
| K-mer model ($k = 6$) | N/A | 29.57 | N/A |
| PhyloP (6D) | N/A | 36.62 | N/A |

Table A3: UCNE identification task: binary classification of UCNEs versus length-matched upstream flanking regions. Accuracy is reported as balanced accuracy. Models in orange are our models pre-trained on sequence alone, models in blue are our models pre-trained to predict PhyloP scores + sequence, and models in purple are pre-trained to predict PhyloP scores alone, with BiGamba MLM+MEM showing the largest gain over its sequence-only counterpart. Percent improvement over non-evolutionary rate pretrained model is included in brackets

| Model | Accuracy (%) |
|---|---|
| ARGAMBA NTP-ONLY | 63.9 |
| ARGAMBA CEP-ONLY | 77.3 |
| ARGAMBA NTP+CEP | **77.6** (+13.7) |
| BIGAMBA MLM-ONLY | 66.2 |
| BIGAMBA MEM-ONLY | **76.0** |
| BIGAMBA MLM+MEM | **81.7** (+15.5) |
| PHYLOGPN | 75.4 |
| NUCLEOTIDE TRANSFORMER (MULTI-SPECIES) | 74.6 |
| NUCLEOTIDE TRANSFORMER (HUMAN-REF) | 74.9 |
| HYENADNA | 49.1 |
| CADUCEUS | 71.0 |

Table A4: Variant effect prediction (VEP) on ClinVar. We report AUCs for log-likelihood and predicted conservation scores, as well as Pearson correlation with true conservation scores where applicable. Models in orange are our models pre-trained on sequence alone, models in blue are our models pre-trained to predict PhyloP scores + sequence, and models in purple are pre-trained to predict PhyloP scores alone.

| Model | Log-likelihood AUC | Pred. cons. AUC | Cons. corr. (Pearson) |
|---|---|---|---|
| ARGAMBA NTP-ONLY | 0.501 | – | – |
| ARGAMBA CEP-ONLY | – | 0.677 | 0.358 |
| ARGAMBA NTP+CEP | 0.512 | 0.696 | 0.398 |
| BIGAMBA MLM-ONLY | 0.501 | – | – |
| BIGAMBA MEM-ONLY | – | 0.737 | 0.455 |
| BIGAMBA MLM+MEM | 0.528 | **0.740** | **0.463** |
| CADUCEUS | 0.640 | – | – |
| HYENADNA (1M SEQ-LEN) | 0.501 | – | – |
| NUCLEOTIDE TRANSFORMER (MULTI-SPECIES) | 0.597 | – | – |
| NUCLEOTIDE TRANSFORMER (HUMAN) | 0.510 | – | – |
| **PhyloP Score** | – | 0.9121 | – |
| **PhyloGPN** | – | 0.960 | – |

