# OpenReview forum: "Predicting evolutionary rate as a pretraining task improves genome language models"
_ICLR.cc/2026/Conference — ICLR 2026 Conference Withdrawn Submission_

### Official Review · Reviewer_GRwb · 2025-10-27

**Soundness:** 2
**Presentation:** 3
**Contribution:** 3
**Rating:** 4
**Confidence:** 3

**Summary:**

The paper proposes evolutionary-rate prediction as an alternative pretraining task for GLMs.
There are two methods: Current Evolution Prediction (CEP) and Masked Evolution Modeling (MEM).
The authors implement these tasks based on Jamba and Caduceus.
The results suggest that explicitly modeling evolutionary conservation provides a strong self-supervised signal for genomic representation learning.

**Strengths:**

(i) Novelty: The paper proposes evolutionary-rate prediction as an alternative pretraining task for GLMs.

(ii) Experimental results: The paper uses two models based on Jamba and Caduceus to validate their methods.

(iii) Benchmark: part of the benchmarks are novel and proposed by the paper.

**Weaknesses:**

(i) The limitation of model architectures: The paper only validates the method based on Jamba and Caduceus GLMs.

(ii) Limitation of the dataset: The paper only considers the human reference genome.

(iii) Limited baseline: The paper uses NT, K-mer methods as baselines, but without considering diverse genomic models.

(iv) Typos: "??" in line 415.

**Questions:**

(i) Whether the method works on the BERT and GPT architecture?

(ii) Whether the method works beyond the human reference genome?

(iii) When I get the evolutionary rate of the DNA sequence to do pretraining, if I only know the DNA sequence, is this enough to get the evolutionary rate information? If so, how to achieve this?

(iv) How about other genomic models' performance on the benchmark?

**Details Of Ethics Concerns:**

I do not have ethical concerns.

---

### Official Review · Reviewer_1pAm · 2025-10-30

**Soundness:** 2
**Presentation:** 2
**Contribution:** 2
**Rating:** 4
**Confidence:** 4

**Summary:**

The authors proposed new pretraining objectives for genomic language model (gLM) pretraining—predicting evolutionary rates estimated by PhyloP based on whole-genome alignments. The objectives can be used on their own or combined with conventional MLM or NTP objectives. I find the idea to be simple but potentially effective, but to fully demonstrate its potential, more comprehensive evaluations are necessary. I therefore recommend a weak rejection in its current form.

**Strengths:**

1. It is an interesting and promising idea to use evolutionary rate prediction as a pretraining task for gLM.
2. The evaluation on zero-shot VEP and UCNE classification showed improved performance compared with gLMs trained on conventional objectives.

**Weaknesses:**

1. My main concern is that the evaluation tasks are limited. I personally do not find the genomic region classification to be a meaningful quantitative evaluation. The authors should include evaluations on some more meaningful benchmarks from BEND (https://openreview.net/forum?id=uKB4cFNQFg), DART-Eval (https://openreview.net/forum?id=qR0x6H5WUX), LRB (https://openreview.net/forum?id=Cdc90HKs1I), etc.
2. Comparison with PhyloGPN should be emphasized more, given its high similarity to the proposed framework in both the concept and the purpose. It appears that PhyloGPN achieves much better zero-shot VEP performance, but worse transfer learning performance on UCNE classification. This should be highlighted and discussed in the main text. And evaluation on more transfer learning tasks is needed to fully demonstrate the superiority of the proposed strategy.

**Questions:**

1. Were both forward and reverse strands on hg38 used for training, or only the forward strand?
2. Ideally, Evo 2 should be added to the comparisons. What was the reason to have not included it?

---

### Official Review · Reviewer_AMk2 · 2025-10-30

**Soundness:** 1
**Presentation:** 2
**Contribution:** 2
**Rating:** 2
**Confidence:** 5

**Summary:**

Can predicting evolutionary rate of each nucleotide in addition to predicting nucleotide itself improve genomic foundational models?

**Strengths:**

1. The idea overall is both novel and important
2. Using some strong baselines and randomly initialized models

**Weaknesses:**

1. **Results**

a. Genomic elements classification is generally too easy, but balanced accuracy seems low. Can you run some other classifier which produces probability estimates and check AUC, micro/macro F1 scores, etc? And you can compare it with results in other papers, such as HyenaDNA.

b. There are mentions of NT benchmark, however, ArGamba and BiGamba were not evaluated on that.

c. Only one pathogenicity prediction task. GPN-MSA and BioFM evaluate common vs rare or non-existent variant classification performance. For example, you can check if your model is able to classify MAF > 5% from MAF<0.001% variants in held-out chromosomes. There are other variant effect tasks, eQTL, sQTL, etc. You mention somewhere that evolution scores are the best variant effect predictor, but test your models ONLY on one pathogenicity prediction task. Also, I am concerned that ground truth PhyloP scores are so much better than even BiGamba with MEM. Given that you used whole pathogenicity dataset (variants from all chromosomes), proper pretraining should be able to memorize most of the PhyloP scores thus making BiGamba with MEM or ArGamba with CEP very close to PhyloP scores.

d. Evaluation on ultra-conserved elements is interesting; but it is even more interesting to see if it can be done in zero-shot fashion. For example, what are the predicted PhyloP score averages and nucleotide likelihoods for these elements?

e. In general, I see that NTP+CEP and MLM+MEM models are better than vanilla ones, albeit only on three tasks out of roughly 50-100 tasks available from many genomic benchmarks out there. But, CEP and MEM utility can be much more obvious if you use it for pretraining or finetuning other, known models, for example Caduceus, which should be easy to replicate/finetune, given your expertise with pretraining mamba-like models.  It will help disentangle influence of your specific pretraining setup and influence of additional pretraining tasks. Because it is not obvious if your NTP or MLM models are bad and you recover some performance using CEP and MEM tasks, or NTP/MLM models are good already in their weight category and you improving its peak performance which cannot be improved by adding more pretraining compute/tuning learning rates/etc. If you can consistently demonstrate that your baseline models are better/competitive with other models on a few benchmarks, I will withdraw this objection.

2. **Pretraining and ablations**

a. I would like to see some pretraining curves, cross-entropy loss and phylop prediction loss on held-out chromosomes during training.

b. It is not obvious that a simple sum of two losses it the optimal approach, maybe PhyloP loss contributes 95% during training, also are you implementing masking of positions where PhyloP scores were not available or use zeros during loss calculation?

c. Why use 4M and 66M parameter models, if you are trying to compare them with Caduceus or Nucleotide Transformer. If you want to rigorously compare them with Caduceus, use 6.6M exactly, if with NT, use <=500M but show that your model is better on average/for some meaningful portion of eval tasks/performs the same, but inference is cheaper.

d. Why bother with Mamba layers if you use 2K context where presumably FlashAttention kernels will be faster and more effective, especially considering modest model sizes?

**Questions:**

Paper does not make a sufficiently strong case that adding evolutionary rate prediction can improve GFM results for a broad range of downstream tasks. One variant effect prediction task and two genomic element classification tasks are definitely not enough to make this case.

I am willing to change my decision if:

1. More variant effect evaluations, some of Dart, TraitGym, VariantBenchmark (BioFM), ProteinGym, MaveDB, AlphaGenome/Borzoi eQTL and sQTL datasets will consistently show that CEP and MEM tasks are helpful across the board or, even better, allow ArGamba and BiGamba beat the newest 2025 generation of GFMs. At least one of these benchmarks have to be there.

2. Use GUE, NT benchmark, NT long-range benchmark, BEND, etc to compare ArGamba and BiGamba with other GFMs. At least one of these benchmarks have to be there.

3. Proper evaluation of how well you memorize PhyloP scores for pretraining dataset and predict them for held-out dataset. If you can predict them in held-out dataset well, it is VERY impressive on its own and can indeed lead to some interesting biological discoveries.

**Feedback**

1. Very hard to understand Figure 2B and 2C. Make points several times larger, name both bioinformatics baselines on 2B, expand Figure captions, 2B caption starts from lower letter.
2. Figure 3A – how exactly are you using PhyloP score? X axis model names on Figure 3C are too verbose, you already use color and Ar/Bi prefixes for most of this information.
3. One link to a supplementary table is missing, around line 415 on page 8.

---

### Official Review · Reviewer_BMst · 2025-10-31

**Soundness:** 2
**Presentation:** 2
**Contribution:** 2
**Rating:** 2
**Confidence:** 3

**Summary:**

This work provides a principled study of the effectiveness of predicting evolutionary rate as a pretraining task. The authors introduce two novel evolutionary rate prediction pre-training tasks. In current evolution prediction (CEP), the model learns to predict the evolutionary rate at each position given the sequence up to that position. In masked evolution modeling (MEM), the model learns to predict evolution rates at masked positions from the surrounding nucleotides.

**Strengths:**

1. The authors investigate alternative pre-training objectives specifically designed for genome modeling, which tries to incorporate biological signals into pretraining.
2. The proposed objective could be combined with previous self-supervision pretext tasks

**Weaknesses:**

1. The task of predicting evolutionary rate is not a self-supervision task and requires estimated evolutionary rate labels, which may limit its further scaling, cause bias towards function prediction and hinder further generalizability to other tasks.
2. The comparison to more SOTA methods should be included.
3. The effectiveness over multiple common model structures should be presented.

**Questions:**

1. In Table A2, the performance of HyenaDNA drops after pretraining, which is uncommon for correct pre-training. Which objectives are used for the pre-training? Why performance drop? and even with performance drop HyenaDNA is still the best model, why not try out pre-training HyenaDNA with the proposed objective?
2. Single-nucleotide resolution may not be the most appropriate resolution, did the authors try out predicting a short sequence of DNA to capture the more general patterns?
3. Instead of predicting evolution rate, will directly ‘pertaining’ on functional region prediction task achieve similar performance? As in principle, both tasks convey very similar biological priors, and are worth further comparing.

---

### Note · Authors · 2025-11-20

I have read and agree with the venue's withdrawal policy on behalf of myself and my co-authors.